# Retinal Photoreceptors and Microvascular Changes in the Assessment of Diabetic Retinopathy Progression: A Two-Year Follow-Up Study

**DOI:** 10.3390/diagnostics13152513

**Published:** 2023-07-27

**Authors:** Magdalena Kupis, Zbigniew M. Wawrzyniak, Jacek P. Szaflik, Anna Zaleska-Żmijewska

**Affiliations:** 1Department of Ophthalmology, SPKSO Ophthalmic Hospital, Medical University of Warsaw, 02-097 Warsaw, Poland; 2Faculty of Electronics and Information Technology, Warsaw University of Technology, 00-665 Warsaw, Poland

**Keywords:** diabetic retinopathy, adaptive optics, rtx-1 technology, cone morphology, retinal microcirculation

## Abstract

Background: With the increasing global incidence of diabetes mellitus (DM), diabetic retinopathy (DR) has become one of the leading causes of blindness in developed countries. DR leads to changes in retinal neurons and microcirculation. Rtx1^TM^ (Imagine Eyes, Orsay, France) is a retinal camera that allows histological visualisations of cones and retinal microcirculation throughout the DM duration. Objective: This study aimed to analyse the cones and retinal microvascular changes in 50 diabetic individuals and 18 healthy volunteers. The patients participated in the initial visit and two follow-up appointments, one and two years after the study, beginning with Rtx1^TM^ image acquisition, visual acuity assessment, macular OCT scans and blood measurements. Results: The study revealed significant differences in the cone density, mosaic arrangement and vascular morphology between healthy and diabetic patients. The final measurements showed decreased photoreceptor and microvascular parameters in the DR group compared with the control group. Furthermore, in the 2-year follow-up, both groups’ Rtx1^TM^-acquired morphological changes were statistically significant. Conclusions: Rtx1^TM^ technology was successfully used as a non-invasive method of photoreceptors and retinal vasculature assessment over time in patients with diabetic retinopathy. The study revealed a trend toward more vascular morphological changes occurring over time in diabetic patients.

## 1. Introduction

Diabetes mellitus (DM) is a chronic metabolic disease affecting multiple tissues with subsequent organ failure. The medical world is struggling with an epidemic of diabetes. According to an epidemiologic study from 2019, there were approximately 463 million DM patients globally, which is expected to increase to 700 million patients in the next 25 years [1]. Diabetic retinopathy (DR) is one of the most common complications of DM and has become one of the leading causes of blindness in developed countries. The development and progression of DR is related to the type and duration of diabetes, glucose levels and blood pressure [2].

DR can be classified into five stages: the first stage of no apparent retinopathy, the second stage of mild non-proliferative diabetic retinopathy (M-NPDR), the third stage of moderate NPDR, the fourth stage of severe NPDR (s-NPDR) and the fifth stage of proliferative diabetic retinopathy (PDR) [3].

Persistent hyperglycaemia and hypoxia results in microvascular changes such as loss of pericytes, basement membrane thickening, arteriole wall thickening and formation of microaneurysms, which can characterise the DR onset. However, multiple studies demonstrated that abnormal neuroretinal function precedes the development of these hallmark vascular lesions [4]. Early perturbations in the diabetic retina involve thinning of the inner retinal layers. Both hyperglycaemia and hypoxia lead to apoptosis of several populations of retinal cells, including retinal ganglion cells, photoreceptors and bipolar cells [5]. Neuroretinal changes may manifest as colour vision defects, decreased contrast sensitivity and electrophysiological abnormalities [6].

As DR is a disease of changes in the retinal neurons and microcirculation, perfect diagnostic procedures should allow early and precise detection. The gold standard for visualising human retinal vessels includes fluorescein angiography (FA). However, early microvascular abnormalities such as arteriole wall thickening are not detectable in FA. Novel optical technologies permit early detection of both microvascular and neuroretinal changes. The abnormal neuroretinal function may be visualised in spectral domain optical coherence tomography (SDOCT) as thinning of the inner retinal layers [7]. Optical coherence tomography angiography (OCTA) enables high-resolution retinal vasculature imaging; however, disrupted segmentation in the retinal layers may affect the visualisation of the vascular flow.

Adaptive optics (AO) technology is an innovative method that allows non-invasive visualisation and quantification of microcirculation and retinal cells in healthy eyes and eyes with retinal diseases. AO has revolutionised the ways of examining eye structures in vivo. It improves the quality of obtained images by compensating for the wavefront aberrations in the eye with a system of deformable mirrors [8]. A series of electric actuators connected to the mirror deform its surface to modify the light beam and thus effectively remove optical distortion in real-time. AO itself does not create an image; this system must be integrated into existing retinal imaging devices such as fundus camera (FC) or OCT. The first device integrated with the AO to visualise the photoreceptor mosaic is FC- AO- and the first commercially available system is rtx1^TM^ (Imagine Eyes, Orsay, France). Multiple studies have used AO technology in diabetic patients. In 2021, Ueno et al. [9] worked on laser speckle flowgraphy and adaptive optics to prove that retinal vessel wall thickening led to a narrowing of the lumen diameter and a decrease in the blood flow in the PDR group. Zaleska-Zmijewska et al. [10] 2019 demonstrated changes in cone density, regularity and the retinal artery parameters between DR and healthy patients.

Rtx1^TM^ is an AO retinal camera using infrared illumination (wavelength 850 nm) with a resolution of 3.5 μm and a field of view of 4 × 4 degrees [11]. The area corresponds to an approximately 1.2 × 1.2 mm^2^ square on the retinal surface. The image acquisition time lasts 4 s and permits the capture of 40 individual images [10]. Compared with other diagnostic technologies, the rtx1^TM^ microscope visualises single retinal cells (photoreceptors) and the minor blood vessel structure with the highest possible resolution. The device permits the acquisition of images in any retinal region and the technology software allows for repeated measurements in the same spots. AOdetect (for the photoreceptors analysis) and AOdetectArtery (for the retinal microvasculature analysis) are two computer programs supplied by the manufacturer to evaluate the rtx-1 retinal spotting.

In the previously published literature, multiple studies have analysed rtx-1 images in diabetic patients [6,9,10,11]. However, as far as we know, no publications have used rtx1^TM^ technology to assess both cone and retinal microvascular parameters over time in patients with diabetes.

Thus, this study aimed to evaluate the cone density and spacing and retinal vascular changes over a prespecified time of DR duration.

## 2. Materials and Methods

It was a prospective study, conducted between May 2018 and February 2021 at the Department of Ophthalmology, Faculty of Medicine, the Medical University of Warsaw, located in the Ophthalmic University Hospital in Warsaw. The Bioethical Commission approved the study protocol of the Medical University of Warsaw (approval number KB/87/2015). All investigations adhered to the tenets of the Declaration of Helsinki. All of the study subjects signed a written informed consent form.

### 2.1. Participants

Patients of the study group were recruited during regular retinal clinical appointments. They were enrolled in the study if the following inclusion criteria were met: age 25–75 and confirmed diagnosis of type-1 or type-2 diabetes mellitus.

The exclusion criteria involved best corrected visual acuity (BCVA) less than 0.5, media opacities to preclude good image quality, no central fixation, history of ocular trauma, history of anti-VEGF (vascular endothelial growth factor) injections, myopia >6 dioptres or astigmatism >2.50 dioptre cylindrical. Subjects with macular pathologies other than diabetic disease were also excluded from the study.

Participants were divided into two groups: diabetic (type-1 or type-2 diabetes mellitus) and control (healthy subjects). The study group consisted of patients with non-proliferative diabetic retinopathy (NPDR), classified as the third stage of moderate NPDR according to the International Clinical Disease Severity Scale for DR [3]. The control group consisted of healthy patients of the same age and with the same exclusion criteria as the diabetic group.

Most participants (72%) from the study group had a history of laser photocoagulation; however, laser treatment did not involve the perifoveal retinal area.

An initial group of 50 diabetic patients previously admitted to the laser service at the Ophthalmic Hospital were enrolled in the study between May 2018 and February 2019. The initial diabetic group (DR) included 31 men (62%) and 19 women (38%). All the patients were scheduled to participate in the initial visit and two follow-up appointments one and two years after the study onset. A total of 10 out of 50 subjects did not attend the last control: 2 died, 1 was currently at the surgery ward diagnosed with diabetic foot syndrome and 7 participants resigned due to the COVID-19 pandemic.

The control group consisted of 18 healthy volunteers, all recruited from hospital staff members, including 3 men (16.6%) and 16 women (83.4%).

### 2.2. Examination Protocol and Parameter Evaluation

The initial appointment included ocular biometry measurements using IOL Master 700 (Carl Zeiss Meditec AG, Hennigsdorf, Germany). Each visit consisted of a comprehensive ophthalmic examination, including refraction and best-corrected visual acuity (BCVA), check (decimal notation), dilated fundus examination, OCT macula scanning (3D DRI OCT Triton) with central retinal thickness (CRT) evaluation and AO retinal image acquisition. The study protocol involved evaluating only one of the subjects’ eyes, usually the one with better visual acuity.

AO camera- Rtx1^TM^ (Imagine Eyes, Orsay, France) was used to acquire images of the four perifoveal regions 2° (approximately 540–600 μm) temporally, nasally, superiorly and inferiorly from the fovea centre. The manufacturer set the standardised sampling window size at 80 μm × 80 μm [12]. In cases where high-quality image acquisition was unobtainable, the pupil was dilated by 1% tropicamide (Polfa, Warszawa). Seven patients (14%) from the study group required pupil dilation, whereas none of the control group participants required 1% tropicamide administration.

For the best possible quality accomplishment, we made 3–4 scans of cone regions and arterioles. Frames with significant motion artefacts caused by blinking or eye movements were excluded. Proprietary programs—AOdetect and AOdetectArtery—were provided by the manufacturer to correct distortions within frames and average the frames for the final image production [8].

In photoreceptor visualisation, the software computed the mean cone density per square millimetre of the retinal surface. It analysed the spatial distribution of the detected cells, including intercell spacing and the number of nearest neighbours, using classical Delaunay triangulation and Voronoi tessellation algorithms [11]. The software calculations included the percentage of cone tailing, the optimal hexagonal (*n* = 6) tailing and 4-, 5-, 7-, 8-, >9 tailing.

Retinal vessel measurements included arterioles located temporally—superiorly or inferiorly—with sizes between 70 and 130 μm. As rtx1^TM^ technology enables precise retinal localisation, we analysed the same vessels in the initial and follow-up appointments. We obtained the vessel diameter (VD), the lumen diameter (LD) and the wall thickness (WT). VD was calculated as the sum of the single arteriolar wall (WT) plus vessel lumen (LD) and single arteriolar wall thickness (WT): VD = WT + (WT + LD). The wall-to-lumen ratio (WLR) was automatically calculated as WLR = 2 × WT/LD; VD and LD usage evaluated the cross-sectional area of the vascular wall (WCSA).

Photoreceptor and retinal parameters were measured in 3 scans; the arithmetic mean of these three values was included in the statistical analysis. We tried eliminating retinal vessels and haemorrhages in the photoreceptor region of choice. The vascular images were recorded at 0.5–1-disc diameter from the optic disc margin, avoiding artery–venous crossings and retinal vein neighbourhoods.

### 2.3. Statistical Analyses

We obtained four measurements of the perifoveal regions located temporally, nasally, superiorly and inferiorly from the fovea in each investigated eye. The data analysis was performed using Dell Inc. (Austin, TX, USA, 2016) Dell Statistica (data analysis software system), version 13.1. (software.dell.com). Continuous variables, expressed as means ± SD (standard deviation), were compared between the DR and control groups using the Student’s *t*-test (*t*-test) or the Mann–Whitney (M-W) U-test, depending on the distribution pattern. The Shapiro–Wilk (S-W) test was used to confirm or reject the normal distribution of each continuous variable. Relationships between numerical variables were analysed using Pearson correlation analysis when the parametric test condition was met and Spearman correlation analysis when the parametric test condition was not met. The *p*-value was based on the two-sided test, with statistical significance considered as *p* < 0.05.

## 3. Results

The study protocol involved evaluating only one of the subjects’ eyes, usually the one with better visual acuity. The mean (±standard deviation) age in the study group was significantly higher than in the control group: 49.7 ± 11.1 vs. 41.6 ± 11.5 (*p* = 0.011 *t*-test; see Table 1). The mean axial length in the DR and the control groups were not significantly different: 23.2 mm ± 1.0 mm for the right and the left eye in the DR group vs. 23.0 mm ± 1.0 mm for the right eye and 22.8 mm ± 1.3 mm for the left eye in the control group (*p* = 0.612 *t*-test for the right eye and *p* = 0.339 *t*-test for the left eye; Table 1). The mean best corrected visual acuity (BCVA) of the investigated eye was significantly lower in the DR group than in the controls in the initial 0.866 ± 0.2 vs 1.0 ± 0 and final examination 0.828 ± 0.173 vs. 0.983 ± 0.038, respectively (M-W test *p* = 0.002 initially and *p* = 0.003 finally; Table 1).

The mean BMI (body mass index) significantly differed between the DR and the control groups in the initial 27.4 ± 4.9 vs. 24.2 ± 1.6 (M-W test *p* = 0.003; Table 1) and final acquisition 27.1 ± 4.4 vs. 24.8 ± 1.7 (M-W test *p* = 0.026; Table 1). Other parameters, including glycated haemoglobin A1c (Hba1c), fasting plasma glucose (FPG) levels, central retinal thickness (CRT) or diagnosed hypertension, are presented in Table 1.

According to the figures presented in Table 1, the initial and final parameters were compared in each group (Table 1). In a 2-year follow-up, statistically relevant changes included a higher percentage of diagnosed hypertension in the DR group and an increase in the mean BMI in the control group (Table 1).

### 3.1. Cone Parameters

The mean initial cone density was significantly lower in the DR group in comparison with the control group in all four retinal locations: 20,380 ± 4652 vs. 25,652 ± 2513 in the temporal, 20,203 ± 4775 vs. 25,165 ± 3261 in the nasal, 19,680 ± 4902 vs. 24,059 ± 2846 in the superior and 19,980 ± 4652 vs. 23,481 ± 3959 in the inferior quadrant (Table 2).

For the final visit, the mean cone density was significantly lower in the DR group in comparison with the control group in all four retinal locations: 19,412 ± 5049 vs. 23,920 ± 3037 in the temporal, 19,634 ± 4521 vs. 24,031 ± 2902 in the nasal, 18,491 ± 4841 vs. 22,768 ± 2826 in the superior and 19,047 ± 5270 vs. 21,885± 3992 in the inferior quadrant. The results are presented in Table 2.

In a 2-year follow-up, the mean final cone density significantly decreased in all four locations in both groups (in consecutive quadrants *p* ≤ 0.001; <0.001; <0.001; <0.001 and 0.002; 0.037; 0.043; <0.001, respectively, in the DR and the control group; Table 2).

The mean initial interphotoreceptor spacing (SM) was significantly higher in all quadrants in the DR group compared with the controls (*p* < 0.05; Table 3). According to Table 3, the mean final SM difference between both groups was statistically relevant apart from the inferior quadrant.

In a 2-year follow-up, the mean interphotoreceptor spacing (SM) increased in the examined groups in all four retinal locations (in the consecutive quadrants *p* ≤ 0.001; 0.002; <0.001; 0.002 and 0.032; 0.606; <0.001; 0.208, respectively, in the DR and the control group; Table 3).

According to Table 4, when comparing the DR group and the controls, there was a significant difference between the cone regularity in the temporal quadrant in the initial trial and the nasal quadrant in the final test (*p* < 0.05). Moreover, in the 2-year observation, the mean cone regularity significantly decreased in most of the retinal locations in the DR group (*p* < 0.05, Table 4).

The regularity of cones was additionally assessed by the mean percentage of Voronoi tiles (N%6), with significantly lower results initially in the temporal, nasal and superior quadrant in the DR group compared with the controls (*p* < 0.05; Table 5). However, in the final trial, Voronoi tile values significantly differed in both groups in the nasal and superior location (*p* < 0.05; Table 5).

Compared with the initial acquisition, the mean final percentage of hexagonal cones was significantly lower in all four retinal locations in the DR group, whereas only in the temporal quadrant in the control group (*p* < 0.05; Table 5).

### 3.2. Retinal Artery Parameters

The mean initial lumen and the total diameter of the analysed retinal artery did not significantly differ between groups (*p* > 0.05; Table 6), even though initially the mean of both artery walls (wall 1 and wall 2) was thicker and the mean WLR and WCSA values were significantly higher in the DR group compared with the controls (*p* < 0.05; Table 6).

In the 2-year follow-up, the increase in WCSA and WLR and the mean value of artery walls was observed in the DR group; nevertheless, a statistically significant difference concerned only the mean value of artery wall 1 (14.5 ± 2.5 vs. 15.1 ± 2.3 (*p* = 0.009)) and wall 2 (14.6 ± 2.4 vs. 15.1 ± 2.3 (*p* = 0.025)) in the initial and final acquisition, respectively (Table 6). The control group observation revealed significant differences in the retinal artery measurements, including increases in the mean value of artery walls and WCSA and WLR parameters (see Table 6). However, the final WLR measurements of the controls remained within the normal range.

Figure 1 and Figure 2 were taken from a DR patient in the initial part of the study and during a 2-year follow-up. Images present lower cone density, regularity and higher interphotoreceptor spacing (SM) after the follow-up period. Microvascular changes shown in Figure 2 include thickening of the arteriole walls and increased WLR and WCSA in the 2-year observation time.

### 3.3. Visual Acuity, Hba1c, FPG, CRT, Age, BMI Correlations

The relationships between changes in cone and retinal artery parameters and factors such as visual acuity, Hba1c, FPG, CRT, age and BMI were determined with Pearson correlation coefficients in the DR group (Table 7). There was a correlation between BCVA deterioration and interphotoreceptor spacing (SM) changes (cone density decreases r = 0.6200 and r = −0.6779, respectively). Both parameters’ changes correlated with age (r= −0.4639 for cone density and r = 0.4014 for cone spacing (SM)). In contrast, other factors, such as CRT, Hba1c and BMI, presented no apparent correlation with the cone or retinal artery measurements.

## 4. Discussion

This study used the rtx-1 AO fundus camera to assess and compare cone density and retinal microvasculature in the 2-year follow-up period in diabetic and healthy volunteers. The study included central retinal thickness measurements, BCVA, BMI evaluation and blood parameters analysis.

Our findings indicated that cone density was significantly lower in the DR group than in the control group in all four retinal quadrants. The research confirmed the previous studies’ results [10,11,13,14,15,16], indicating a difference in the photoreceptor counts between healthy and diabetic patients. Lammer et al. [6] proved no relevant change in the cone density and spacing in DR; however, their study emphasised a consistent association between the regularity of cone arrangement and DR presence.

We did not observe significant differences in cone parameters in the DR group in all four retinal locations. These results were similar to the findings of multiple previous studies [10,13,17,18]. In contrast with our findings, Cristescu et al. 2019 proved an asymmetry between the cone density in horizontal and vertical meridians in DR and the control group [14]; the authors indicated that the higher density of cones in the horizontal line might be explained by the vision usage while reading. Moreover, in our study, the control group presented a variability of cone density in four meridians (Table 2). In line with our findings, Zaleska-Zmijewska et al. and Park et al. stated that the highest density of cones among healthy patients was in the temporal quadrant [17,19]. Further studies must determine this issue because of the discrepancy between cone densities across retinal meridians in diabetic and healthy subjects.

Our study revealed increased interphotoreceptor spacing distance, decreased regularity and cone rearrangement mosaic (evaluated by the percentage of hexagonal cones) in DR patients compared with controls. Previous analyses indicated the same results [6,10,13]. Lammer et al. suggested that the cone regularity and mosaic changes may result from retinal cell swelling in diabetic macular oedema (DME) [6]. As DME is related to photoreceptor layer disruption [20], the history of DME may cause cone density loss. However, our study eliminated a possible influence of DME on cone mosaic by excluding participants with previous anti-VEGF treatment. Apart from macular oedema, cone spacing and regularity may be affected by retinal pathologies such as intraretinal cysts, extracellular fluid accumulation, haemorrhages and hard exudates that may commonly appear in diabetes [10].

It is commonly known that microvascular architecture is affected by glucose levels and blood pressure. In our study, in the microvasculature analysis, we observed thicker artery walls, higher WLR and higher WCSA in diabetic patients than in healthy ones, with no significant difference in the lumen and external diameter. Similar results were presented by Ueno et al., suggesting that the stiffening of the collagen may cause vascular wall changes in diabetic patients due to advanced glycosylated products with collagen cross-linking and by the growth of the muscle cells [9]. Our study involved an inquiry about coexisting hypertension, but further examination using AO technology should be performed to assess blood pressure influence on vascular changes in diabetic patients.

No other studies have analysed both cone and retinal changes over time among DR and healthy patients. After two years of our research, we observed a significantly lower cone density in the DR group compared with the controls. Moreover, we found a lower correlation between cone densities in the retinal quadrants in the diabetic group. It may suggest that the illness duration can be associated with a higher diversity of cone parameters between different retinal locations.

In our 2-year-time study, we detected that an increase in interphotoreceptor spacing cone regularity was significantly lower in all retinal locations in the examined group. As was previously stated, no other observational studies have assessed changes in cone density and arrangement in time; however, some papers have noted no correlation between cone density and the duration of diabetes [6,13]. Our findings indicated lower cone density in healthy and diabetic patients over time. Some authors have found a relationship between cone parameters and age factors, but others have not found such correlations [6,19,21]. Our study found that changes in the cone density measured over time were more minor in the DR group than in the controls. Sex inequality in the DR and the control group was considered a potential reason for the cone changes’ disproportion; nevertheless, multiple studies deny sex influence on cone density [14,17].

Most differences in microvasculature parameters in the 2-year time in our study group were slight and not statistically significant. The results may correspond with stable Hba1c and FPG levels in the DR group, which indicated well-controlled diabetes in our study group. There are no other studies analysing microvasculature retinal changes occurring over time in diabetic and healthy patients, so it was complicated to compare our results. However, studies have proven that the severity of diabetes significantly affects lumen diameters and WLR [9,22,23]. It has been confirmed that photocoagulation leads to retinal vessel narrowing, autoregulatory vasoconstriction and reduced retinal blood flow [24,25]. Moreover, Sugimoto et al. stated that anti-VEGF agents are responsible for ocular blood flow diminishment [26]; thus, our study excluded patients with a history of anti-VEGF treatment. The vascular changes in the control group may have been determined by an increase in BMI, consistent with other previously published studies [12,27].

In the 2-year time analysis, we have proven a significant relationship between BCVA decrease, age augmentation and deterioration of the AO parameters. Our findings align with the results of Cheng et al., who demonstrated greater cone density and smaller spacing in patients with better BCVA [28]. Age factor was already discussed in this article, mentioning various study research opinions. Morphological variables in the diabetic group were not significantly related to HbA1c levels. Our results were similar to those of several other authors [9,10,13]; however, Lombardo et al. [15] and Arichika et al. [23] demonstrated significant relation between cone and vascular morphology with HbA1c changes.

### Study Limitations

This study faced several limitations. First of all, the study size could have been more significant. It was mainly due to the difficulty in recruiting diabetic patients without lens opacification and macular oedema, which can interfere with eye fixation and precise AO image acquisition. Because the study was conducted between 2018 and 2021, the sample size was affected by the COVID-19 epidemic situation. Secondly, the rtx-1 did not allow for assessing the density of tightly packed cones in the centre of the fovea, so it remained questionable whether the perifoveal cone loss corresponded to the centre of the fovea [13]. Finally, the control group was relatively small, with sex and age imbalances, mainly because of restricted exclusion criteria, no history of widespread diseases and the unstable pandemic.

## 5. Conclusions

This study revealed significant differences in mean cone density, mosaic arrangement and microvascular morphology between healthy and diabetic patients. In the 2-year follow-up period, we detected photoreceptor loss and rearrangement in both groups. We found a trend toward vascular morphological changes occurring over time in diabetic patients and controls. To thoroughly compare changes in microvasculature and photoreceptor loss in the duration of the disease, future studies with a larger sample size should be conducted. This study may provide more insight into the DR progression assessment, with the potential for earlier detection of diabetic pathologies.

## Figures and Tables

**Figure 1 diagnostics-13-02513-f001:**
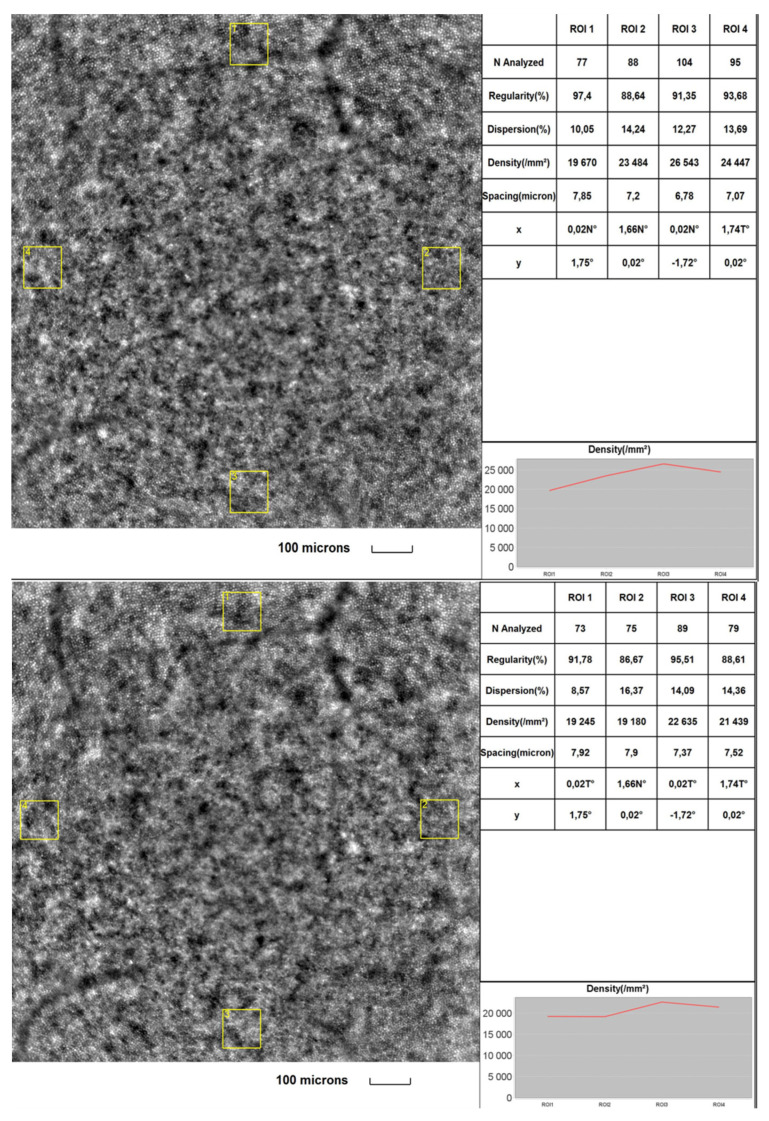
Image of the macular region with four retinal acquisitions, ROI (region of interest) for a DR patient initially (**upper**) and after a 2-year time (**lower**) captured by rtx1^TM^ AO retinal camera. Cone analysis included: regularity (%), dispersion (%), density (cone/mm^2^) and spacing (µm).

**Figure 2 diagnostics-13-02513-f002:**
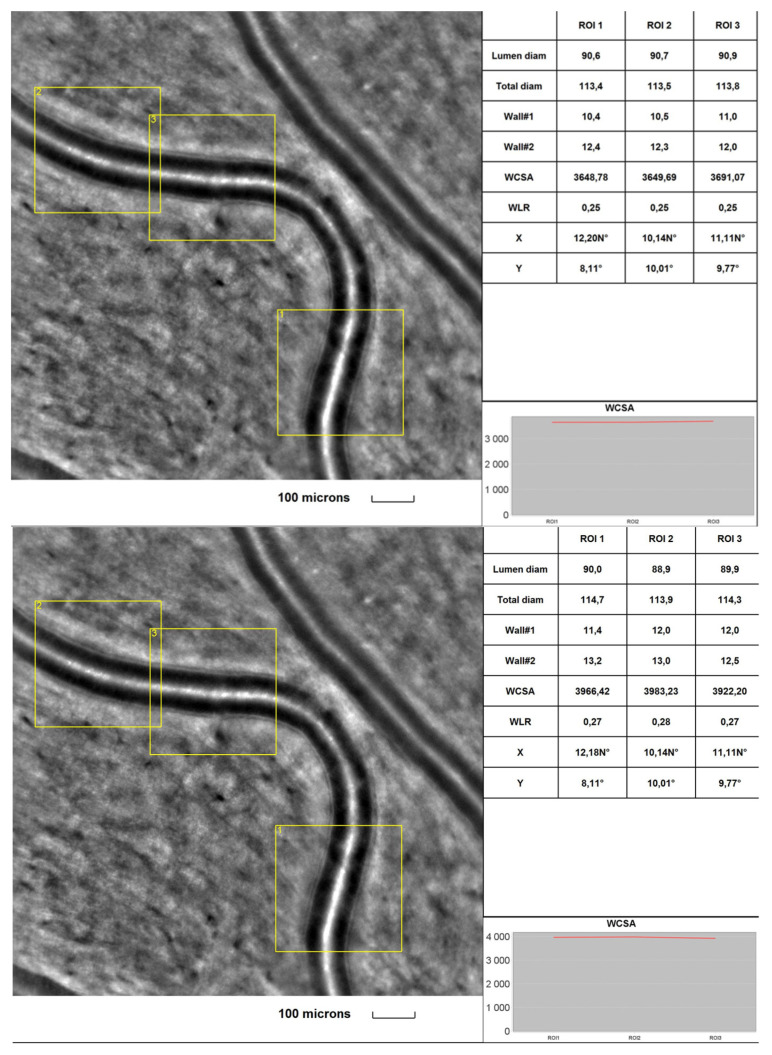
Image of the retinal artery of a DR patient captured by rtx1^TM^ AO retinal camera in three different acquisitions: ROI (region of interest), initially (**upper**) and after a 2-year observation time (**lower**). The charts present the following parameters: Lumen diam—lumen diameter; Total diam—total diameter; wall1 and wall2, WCSA—cross-sectional wall area; —WLR—wall-to-lumen ratio.

**Table 1 diagnostics-13-02513-t001:** Group characteristics and results of the blood parameters.

Characteristics	m ± SD DR	*p*-Value Wilcoxon Signed Rank Test	m ± SD Control	*p*-Value Wilcoxon Signed Rank Test	*p*-Value *t*-Test
Age (years)	49.7 ± 11.1		41.6 ± 11.5		0.011
Initial Hypertension (%)	23.3	0.043	0.0		0.135 ^†^
Final Hypertension (%)	34.9		0.0		0.033 ^†^
AL-R (mm)	23.2 ± 1.0	0.416 ^‡^	23.0 ± 1.0	0.211 ^‡^	0.612
AL-L (mm)	23.2 ± 1.0		22.8 ± 1.3		0.339
Initial BCVA	0.866 ± 0.173	0.233	1.000 ± 0.000	0.109	0.002 ^†^
Final BCVA	0.828 ± 0.203		0.983 ± 0.038		0.003 ^†^
Initial BMI (kg/m^2^)	27.4 ± 4.9	0.200	24.2 ± 1.6	0.006	0.003 ^†^
Final BMI (kg/m^2^)	27.1 ± 4.4		24.8 ± 1.7		0.026 ^†^
Initial Hba1c (%)	8.3 ± 1.8	0.507	5.6 ± 0.6		<0.001 ^†^
Final Hba1c (%)	8.1± 1.8		5.6 ± 0.6		<0.001 ^†^
Initial FPG (mg/dL)	168.5 ± 49.8	0.507	93.4 ± 7.0	0.276	<0.001 ^†^
Final FPG (mg/dL)	162.5 ± 44.7		94.0 ± 4.9		<0.001 ^†^

^†^ *p*-value Mann–Whitney test; ^‡^ *p*-value paired *t*-test; m—mean; SD—standard deviation; AL-R axial length of the right eye; AL-L—axial length of the left eye; BCVA—best-corrected vision acuity; BMI—body mass index; Hba1c—glycated haemoglobin A1c; FPG—fasting plasma glucose.

**Table 2 diagnostics-13-02513-t002:** Initial and final cone density in 4 retinal locations in the DR and the control groups.

Quadrants	Initial Visit			Final Visit			*p*-Value Paired *t*-Test DR	*p*-Value Paired *t*-Test Control
	MCD (± SD) (cone/mm^2^) DR	MCD (±SD) (cone/mm^2^) Control	*p*-Value *t*-Test	MCD (±SD) (cone/mm^2^) DR	MCD (±SD) (cone/mm^2^) Control	*p*-Value *t*-Test		
T	20,380 *** ± 4652	25,652 *** ± 2513	<0.001	19,412 ***± 5049	23,920 *** ±3037	0.001	<0.001	0.002
N	20,203 *** ± 4775	25,165 *** ± 3261	<0.001	19,634 *** ± 4521	24,031 *** ±2902	<0.001	<0.001	0.037
S	19,680 *** ± 4902	24,059 *** ± 2846	<0.001	18,491 *** ± 4841	22,768 *** ± 2826	0.001	<0.001	0.043
I	19,980 ** ± 4652	23,481 ** ± 3959	0.007	19,047 *± 5270	21,885 * ± 3992	0.047	<0.001	<0.001

* *p* < 0.05; ** *p* < 0.001; *** *p* < 0.001; MCD—mean cone density; SD—standard deviation; T—temporal; N—nasal; S—superior; I—inferior.

**Table 3 diagnostics-13-02513-t003:** Initial and final interphotoreceptor spacing in 4 retinal locations in the DR and the control groups.

Quadrants	Initial Visit			Final Visit			*p*-Value Paired *t*-Test DR	*p*-Value Paired *t*-Test Control
	SM (±SD) (µm) DR	SM (±SD) (µm) Control	*p*-Value *t*-Test	SM (±SD) (µm) DR	SM (±SD) (µm) Control	*p*-Value *t*-Test		
T	7.84 *** ± 0.90	6.96 *** ± 0.42	<0.001	8.06 ** ± 1.11	7.25 ** ± 0.56	0.003	<0.001	0.001
N	7.88 *** ± 0.94	7.01 *** ± 0.48	<0.001	7.95 ** ± 0.9	7.24 ** ± 0.63	0.006	0.002	0.160
S	7.99 *** ± 0.98	7.14 *** ± 0.41	0.001	8.24 ** ± 1.1	7.35 ** ± 0.47	0.004	<0.001	0.063
I	7.91 * ± 0.92	7.28 * ± 0.64	0.012	8.11 ± 1.12	7.49 ± 0.71	0.063	0.002	0.010

* *p* < 0.05; ** *p* < 0.01; *** *p* < 0.001; SM—interphotoreceptor spacing; SD—standard deviation; T—temporal; N—nasal; S—superior; I—inferior.

**Table 4 diagnostics-13-02513-t004:** Initial and final cone regularity in 4 retinal locations in the DR and the control groups.

Quadrants	Initial Visit			Final Visit			*p*-Value Paired *t*-Test DR	*p*-Value Paired *t*-Test Control
	Reg (±SD) (%) DR	Reg (±SD) (%) Control	*p*-Value *t*-Test	Reg (±SD) (%) DR	Reg (±SD) (%) Control	*p*-Value *t*-Test		
T	91.2 * ± 3.6	93.3 * ± 3.3	0.042	89.4 ± 5.3	90.7 ± 4.1	0.343	<0.001	0.032
N	90.9 ± 3.8	92.4 ± 4.7	0.212	90.4 * ± 4.2	93.0 * ± 2.8	0.023	0.002	0.606
S	82.3 ± 10.0	85.4 ± 7.0	0.244	91.5 ± 5.2	92.7 ± 4.4	0.419	<0.001	<0.001
I	92.1 ± 4.0	93.8 ± 3.4	0.129	89.9 ± 5.0	92.5 ± 4.9	0.103	0.002	0.208

* *p* < 0.05; Reg—cone regularity; SD—standard deviation; T—temporal; N—nasal; S—superior; I—inferior.

**Table 5 diagnostics-13-02513-t005:** Initial and final percentage of hexagonal cones (6-Voronoi) in 4 retinal locations in the DR and the control groups.

Quadrants	Initial Visit			Final Visit			*p*-Value Paired *t*-Test DR	*p*-Value Paired *t*-Test Control
	6-Voronoi (±SD) (%) DR	6-Voronoi (±SD) (%) Control	*p*-Value *t*-Test	6-Voronoi (±SD) (%) DR	6-Voronoi (±SD) (%) Control	*p*-Value *t*-Test		
T	42.8 * ± 5.1	46.8 * ± 6.9	0.013	40.9 ± 7.0	41.1 ± 6.7	0.946	0.073	0.011
N	44.1 * ± 4.9	44.2 * ± 7.4	0.026 ^†^	39.8 *** ± 6.5	47.6 *** ± 7.8	0.001 ^†^	<0.001 ^‡^	0.233
S	43.8 * ± 6.9	49.0 * ± 13.4	0.020 ^†^	40.1 * ± 8.5	46.9 * ± 9.7	0.011	<0.001 ^‡^	0.979
I	44.0 ± 6.3	45.8 ± 7.1	0.340	41.1 ± 7.7	44.8 ± 9.4	0.160	0.002	0.743

* *p* < 0.05; *** *p* < 0.001; ^†^
*p*-value Mann–Whitney test; ^‡^ *p*-value Wilcoxon signed rank test; 6-Voronoi—a percentage of hexagonal cones; SD—standard deviation; T—temporal; N—nasal; S—superior; I—inferior.

**Table 6 diagnostics-13-02513-t006:** Initial and final retinal artery measurements in 2-year observation time in the DR and the control groups.

Quadrants	Initial Visit			Final Visit			*p*-Value Paired *t*-Test DR	*p*-Value Paired *t*-Test Control
	M (±SD) DR	M (±SD) Control	*p*-Value *t*-Test	M (±SD) DR	M (±SD) Control	*p*-Value *t*-Test		
LD (±SD) (µm)	92.6 ± 18.7	95.9 ± 10.4	0.495	90.4 ± 13.5	95.9 ± 13.2	0.155	0.180	0.495
VD (±SD) (µm)	119.0 ± 17.4	120.0 ± 13.2	0.836	120.8 ± 17.2	120.5 ± 15.1	0.953	0.773	0.836
WALL1 (±SD) (µm)	14.5 *** ± 2.5	12.0 *** ± 1.9	<0.001 ^†^	15.1 *** ± 2.3	12.3 *** ± 1.8	<0.001	0.009 ^‡^	<0.001
WALL2 (±SD) (µm)	14.6 *** ± 2.4	12.0 *** ± 1.5	<0.001 ^†^	15.1 *** ± 2.3	12.6 *** ± 1.7	<0.001	0.025 ^‡^	<0.001
WCSA (±SD) (μm^2^)	4938 * ± 1358	4122 * ± 979	0.033 ^†^	5081 * ± 1254	4268 * ± 978	0.017	0.480 ^‡^	0.033
WLR (±SD)	0.342 *** ± 0.063	0.251 *** ± 0.015	<0.001 ^†^	0.343 *** ± 0.048	0.262 *** ± 0.032	<0.001	0.202 ^‡^	<0.001 ^‡^

* *p* < 0.05; *** *p* < 0.001; ^†^ *p*-value Mann–Whitney test; ^‡^ *p*-value Wilcoxon signed rank test; M—mean; SD—standard deviation; LD—lumen diameter; VD—vessel diameter; WLR—wall-to-lumen ratio; WCSA—cross-sectional wall area.

**Table 7 diagnostics-13-02513-t007:** Correlation between changes in BCVA, CRT, HbA1c, FPG, age, BMI and cone and vascular parameters in the 2-year-time observation using Pearson correlation analysis.

Change between Final and Initial Acquisition	BCVA	CRT	Hba1c	FPG	Age	BMI
MCD	0.6200	−0.0804	0.0804	0.1647	−0.4639	−0.1103
SM	−0.6779	0.1385	−0.0113	−0.0977	0.4014	0.0602
Reg	0.1102	0.0860	−0.0680	−0.0599	−0.3150	−0.0986
6-Voronoi	0.1894	0.0886	0.0752	−0.0465	−0.1154	−0.0792
LD	−0.3572	0.1484	0.0040	−0.1453	0.0978	0.2308
VD	−0.2344	0.1538	0.0189	−0.1970	0.2123	0.2552
WALL1	−0.1344	0.0681	0.1624	−0.0692	0.2504	0.0162
WALL2	0.1387	0.1090	0.2105	−0.1347	0.2309	0.1706
WCSA	−0.2022	0.1559	0.1507	−0.1105	0.2363	0.0960
WLR	0.3389	−0.1664	0.1353	0.0941	−0.0856	−0.0150

BCVA—best-corrected vision acuity; CRT—central retinal thickness; Hba1c—glycated haemoglobin A1c; FPG—fasting plasma glucose; BMI—body mass index; MCD—mean cone density; SM—interphotoreceptor spacing; Reg—cone regularity; 6-Voronoi—the percentage of hexagonal cones; LD—lumen diameter; VD—vessel diameter; WLR—wall-to-lumen ratio; WCSA—wall cross-sectional area.

## Data Availability

The data sets used and analysed during this study are available from the corresponding author in consideration of potentially applying restrictions on reasonable request.

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
