# Peer review of "Retinal Photoreceptors and Microvascular Changes in the Assessment of Diabetic Retinopathy Progression: A Two-Year Follow-Up Study"

_diagnostics, 2023, doi:10.3390/diagnostics13152513_

Round 1

Reviewer 1 Report

Review of the article entitled: ”Retinal photoreceptors and microvascular changes in the assessment of diabetic retinopathy progression – a two-year follow-up study.”

First of all, the article is presented as a review which it is not. This is an original article on a clinical study.

The whole article and technology is difficult to understand for the layman. I remember of many technical and artifacts during a demonstration of the instrument. As a non-user of the technology, the article fails to convince me of the relevance of such a study. The figure does not help either.

The fact that the English is sometimes difficult to understand doesn’t contribute to the clarity of the article:

-        The first sentence of the introduction is awkward

-        Line 69: based is difficult to understand; the authors probably mean depending on..

-        Line 86: it was a retrospective (was it really retrospective?), single centre study with the rtx-1 technology (of course the centre had the technology, obviously no need to mention it)

-        Lines 203-204 difficult to understand (probably a language problem)

-        Line 303 Lammer et al. proved no substantial (awkward)

-        Line 351 “slightly more minor” (change English wording)

-        Many other language problems. Manuscript should be re-written in better English; this might add clarity

In the introduction, in the paragraph reviewing the methods of analysis of retinal vasculature something should be said about OCT-A.

Line 123 (BCVA) check (can be left away) Which technique was used decimal Snellen method?

The authors say they avoided laser scars, but usually perifoveally no laser is performed unless oedema is present but the authors excluded cases with oedema.

The distance of the acquisition of images should also be given in µm (line 128)

In the text the disease group is abbreviated with DR and in the tables it is DM

BMI does not seem a relevant parameter and explaining the vascular chnge in the controls appears as very far-fetched.

Second paragraph of discussion: the divergent findings in the literature cited casts some doubt on the technology.

As it is now, the article is confusing. A whole reshuffle, language-wise and in the presentation of the data, is advisable

The revised manuscript should be submitted to a person familiar with the AO technique

See general comments

Reviewer 2 Report

 The article is an important one of it s category with  relevant
significant differences in the cone density, mosaic arrangement and vascular morphology between healthy and diabetic patients.  The main aim of this article consists into the measurements  of photoreceptor and microvascular parameters in the diabetes  retinopathy group compared to the control group.The study have a consistent group : 50 diabetic individuals and 18 healthy volunteers, with a new tehnics :Rtx1TM (Imagine Eyes, France) - a microscope that allows histological visualisations of cones and retinal microcirculation throughout the diabetes mellitus  duration.

The results are clearly presented  and are in concordance wth the aim of the study :Rtx-1 technology was successfully used as a non-invasive method of photoreceptors  and retinal vasculature assessment over time in patients with diabetic retinopathy.

I thinck that research design was appropriate  with high interest to the readers ., with several limitations of the study .

 My recommandations are to accept the article in present form.

The quality of English is a good one ; the authors have an academic language , well made and playable

Author Response

Dear Reviewer, 

We are very pleased to receive such a positive review. We are grateful for your recommendations to accept the article. 

Yours sincerely, 

Authors